# Robust Transfer Principal Component Analysis with Rank Constraints

**Yuhong Guo**
Department of Computer and Information Sciences
Temple University, Philadelphia, PA 19122, USA
`yuhong@temple.edu`

## Abstract

Principal component analysis (PCA), a well-established technique for data analysis and processing, provides a convenient form of dimensionality reduction that is effective for cleaning small Gaussian noises presented in the data. However, the applicability of standard principal component analysis in real scenarios is limited by its sensitivity to large errors. In this paper, we tackle the challenge problem of recovering data corrupted with errors of high magnitude by developing a novel robust transfer principal component analysis method. Our method is based on the assumption that useful information for the recovery of a corrupted data matrix can be gained from an uncorrupted related data matrix. Specifically, we formulate the data recovery problem as a joint robust principal component analysis problem on the two data matrices, with common principal components shared across matrices and individual principal components specific to each data matrix. The formulated optimization problem is a minimization problem over a convex objective function but with non-convex rank constraints. We develop an efficient proximal projected gradient descent algorithm to solve the proposed optimization problem with convergence guarantees. Our empirical results over image denoising tasks show the proposed method can effectively recover images with random large errors, and significantly outperform both standard PCA and robust PCA with rank constraints.

## 1 Introduction

Dimensionality reduction, as an important form of unsupervised learning, has been widely explored for analyzing complex data such as images, video sequences, text documents, etc. It has been used to discover important latent information about observed data matrices for visualization, feature recovery, embedding and data cleaning. The fundamental assumption roots in dimensionality reduction is that the intrinsic structure of high dimensional observation data lies on a low dimensional linear subspace. Principal component analysis (PCA) [7] is a classic and one of most commonly used dimensionality reduction method. It seeks the best low-rank approximation of the given data matrix under a well understood least-squares reconstruction loss, and projects data onto uncorrelated low dimensional subspace. Moreover, it admits an efficient procedure for computing optimal solutions via the singular value decomposition. These properties make PCA a well suited reduction method when the observed data is mildly corrupted with small Gaussian noise [12]. But standard PCA is very sensitive to the high magnitude errors of the observed data. Even a small fraction of large errors can cause severe degradation in PCA's estimate of the low rank structure.

Real-life data, however, is often corrupted with large errors or even missing observations. To tackle dimensionality reduction with arbitrarily large errors and outliers, a number of approaches that robustify PCA have been developed in the literature, including $\ell_1$-norm regularized robust PCA [14], influence function techniques [5, 13], and alternating $\ell_1$-norm minimization [8]. Nevertheless, the

capacity of these approaches on recovering the low-rank structure of a corrupted data matrix can still be degraded with the increasing of the fraction of the large errors.

In this paper, we propose a novel robust transfer principal component analysis method to recover the low rank representation of heavily corrupted data by leveraging related uncorrupted auxiliary data. Seeking knowledge transfer from a related auxiliary data source for the target learning problem has been popularly studied in supervised learning. It is also known that modeling related data sources together provides rich information for discovering theirs shared subspace representations [4]. We extend such a transfer learning scheme into the PCA framework to perform joint robust principal component analysis over a corrupted target data matrix and a related auxiliary source data matrix by enforcing the two robust PCA operations on the two data matrices to share a subset of common principal components, while maintaining their unique variations through individual principal components specific for each data matrix. This robust transfer PCA framework combines aspects of both robust PCA and transfer learning methodologies. We expect the critical low rank structure shared between the two data matrices can be effectively transferred from the uncorrupted auxiliary data to recover the low dimensional subspace representation of the heavily corrupted target data in a robust manner. We formulate this robust transfer PCA as a joint minimization problem over a convex combination of least squares losses with non-convex matrix rank constraints. Though a simple relaxation of the matrix rank constraints into convex nuclear norm constraints can lead to a convex optimization problem, it is very difficult to control the rank of the low-rank representation matrix we aim to recover through the nuclear norm. We thus develop a proximal projected gradient descent optimization algorithm to solve the proposed optimization problem with rank constraints, which permits a convenient closed-form solution for each proximal step based on singular value decomposition and converges to a stationary point. Our experiments over image denoising tasks show the proposed method can effectively recover images corrupted with random large errors, and significantly outperform both standard PCA and robust PCA with rank constraints.

**Notations:** In this paper, we use $I_n$ to denote an $n \times n$ identify matrix, use $O_{n,m}$ to denote an $n \times m$ matrix with all 0 values, use $\| \cdot \|_F$ to denote the matrix Frobenius norm, and use $\| \cdot \|_*$ to denote the nuclear norm (trace norm).

## 2 Preliminaries

Assume we are given an observed data matrix $X \in \mathbb{R}^{n \times d}$ consisting of $n$ observations of $d$-dimensional feature vectors, which was generated by corrupting some entries of a latent low-rank matrix $M \in \mathbb{R}^{n \times d}$ with an error matrix $E \in \mathbb{R}^{n \times d}$ such that $X = M + E$. We aim to to recover the low-rank matrix $M$ by projecting the high dimensional observations $X$ into a low dimensional manifold representation matrix $Z \in \mathbb{R}^{n \times k}$ over the low dimensional subspace $B \in \mathbb{R}^{k \times d}$, such that $M = ZB$, $BB^\top = I_k$ for $k < d$.

### 2.1 PCA

Given the above setup, standard PCA assumes the error matrix $E$ contains small i.i.d. Gaussian noises, and seeks optimal low dimensional encoding matrix $Z$ and basis matrix $B$ to reconstruct $X$ by $X = ZB + E$. Under a least squares reconstruction loss, PCA is equivalent to the following self-supervised regression problem

$$\min_{Z,B} \|X - ZB\|_F^2 \quad \text{s.t.} \;\; BB^\top = I_k. \tag{1}$$

That is, standard PCA seeks the best rank-$k$ estimate of the latent low-rank matrix $M = ZB$ by solving

$$\min_{M} \|X - M\|_F^2 \quad \text{s.t.} \;\; rank(M) \le k. \tag{2}$$

Although the optimization problem in (1) or (2) is not convex and does not appear to be easy, it can be efficiently solved by performing a singular value decomposition (SVD) over $X$, and permits the following closed-form solution

$$B^* = V_k^\top, \; Z^* = XB^*, \; M^* = Z^*B^*, \tag{3}$$

where $V_k$ is comprised of the top $k$ right singular vectors of $X$. With the convenient solution, standard PCA has been widely used for modern data analysis and serves as an efficient and effective dimensionality reduction procedure when the error $E$ is small and i.i.d. Gaussian [7].

## 2.2 Robust PCA

The validity of standard PCA however breaks down when corrupted errors in the observed data matrix are large. Note that even a single grossly corrupted entry in the observation matrix $X$ can render the recovered $M^*$ matrix to be shifted away from the true low-rank matrix $M$. To recover the intrinsic low-rank matrix $M$ from the observation matrix $X$ corrupted with sparse large errors $E$, a polynomial-time robust PCA method has been developed in [14], which induces the following optimization problem

$$\min_{M,E} \; rank(M) + \gamma \|E\|_0 \quad \text{s.t.} \; X = M + E. \tag{4}$$

By relaxing the non-convex rank function and the $\ell_0$-norm into their convex envelopes of nuclear norm and $\ell_1$-norm respectively, a convex relaxation of the robust PCA can be yielded

$$\min_{M,E} \; \|M\|_* + \lambda \|E\|_1 \quad \text{s.t.} \; X = M + E. \tag{5}$$

With an appropriate choice of $\lambda$ parameter, one can exactly recover the $M, E$ matrices that generated the observations $X$ by solving this convex program.

To produce a scalable optimization for robust PCA, a more convenient relaxed formulation has been considered in [14]

$$\min_{M,E} \; \|M\|_* + \lambda \|E\|_1 + \frac{\alpha}{2} \|M + E - X\|_F^2 \tag{6}$$

where the original equality constraint is replaced with a reconstruction loss penalty term. This formulation apparently seeks the lowest rank $M$ that can best reconstruct the observation matrix $X$ subjecting to sparse errors $E$.

Robust PCA though can effectively recover the low-rank matrix given very sparse large errors in the observed data, its performance can be degraded when the observation data is heavily corrupted with dense large errors. In this work, we propose to tackle this problem by exploiting information from related uncorrupted auxiliary data.

## 3 Robust Transfer PCA

Exploring labeled information in a related auxiliary data set to assist the learning problem on a target data set has been widely studied in supervised learning scenarios within the context of transfer learning, domain adaptation and multi-task learning [10]. Moreover, it has also been shown that modeling related data sources together can provide useful information for discovering their shared subspace representations in an unsupervised manner [4]. The principle behind these knowledge transfer learning approaches is that related data sets can complement each other on identifying the intrinsic latent structure shared between them.

Following this transfer learning scheme, we present a robust transfer PCA method for recovering low-rank matrix from a heavily corrupted observation matrix. Assume we are given a target data matrix $X_t \in \mathbb{R}^{n_t \times d}$ corrupted with errors of large magnitude, and a related source data matrix $X_s \in \mathbb{R}^{n_s \times d}$. The robust transfer PCA aims to achieve the following robust joint matrix factorization

$$X_s = N_s B_c + Z_s B_s + E_s, \tag{7}$$
$$X_t = N_t B_c + Z_t B_t + E_t, \tag{8}$$

where $B_c \in \mathbb{R}^{k_c \times d}$ is the orthogonal basis matrix shared between the two data matrices, $B_s \in \mathbb{R}^{k_s \times d}$ and $B_t \in \mathbb{R}^{k_t \times d}$ are the orthogonal basis matrices specific to each data matrix respectively, $N_s \in \mathbb{R}^{n_s \times k_c}, N_t \in \mathbb{R}^{n_t \times k_c}, Z_s \in \mathbb{R}^{n_s \times k_s}, Z_t \in \mathbb{R}^{n_t \times k_t}$ are the corresponding low dimensional reconstruction coefficient matrices, $E_s \in \mathbb{R}^{n_s \times d}$ and $E_t \in \mathbb{R}^{n_t \times d}$ represent the additive errors in each data matrix. Let $Z_c = [N_s; N_t]$. Given constant matrices $A_s = [I_{n_s}, O_{n_s,n_t}]$ and $A_t = [O_{n_t,n_s}, I_{n_t}]$, we can re-express $N_s$ and $N_t$ in term of the unified matrix $Z_c$ such that $N_s = A_s Z_c$ and $N_t = A_t Z_c$. The learning problem of robust transfer PCA can then be formulated as the following joint minimiza-

tion problem

$$\min_{Z_c, Z_s, Z_t, B_c, B_s, B_t, E_s, E_t} \quad \frac{\alpha_s}{2} \|A_s Z_c B_c + Z_s B_s + E_s - X_s\|_F^2 + \tag{9}$$

$$\frac{\alpha_t}{2} \|A_t Z_c B_c + Z_t B_t + E_t - X_t\|_F^2 + \beta_s \|E_s\|_1 + \beta_t \|E_t\|_1$$

$$\text{s.t.} \quad B_c B_c^\top = I_{k_c}, \ B_s B_s^\top = I_{k_s}, \ B_t B_t^\top = I_{k_t}$$

which minimizes the least squares reconstruction losses on both data matrices with $\ell_1$-norm regularizers over the additive error matrices. The intuition is that by sharing the common column basis vectors in $B_c$, one can best capture the common intrinsic low-rank structure of the data based on limited observations from both data sets, and by allowing data embedding onto individual basis vectors $B_s, B_t$, the additional low-rank structure specific to each data set can be captured. Nevertheless, this is a difficult non-convex optimization problem as both the objective function and the orthogonality constraints are non-convex. To simplify this optimization problem, we introduce replacements

$$M_c = Z_c B_c, \ M_s = Z_s B_s, \ M_t = Z_t B_t \tag{10}$$

and rewrite the optimization problem (9) equivalently into the formulation below

$$\min_{M_c, M_s, M_t, E_s, E_t} \quad \frac{\alpha_s}{2} \|A_s M_c + M_s + E_s - X_s\|_F^2 + \frac{\alpha_t}{2} \|A_t M_c + M_t + E_t - X_t\|_F^2 \tag{11}$$

$$+ \beta_s \|E_s\|_1 + \beta_t \|E_t\|_1$$

$$\text{s.t.} \quad rank(M_c) \le k_c, \quad rank(M_s) \le k_s, \quad rank(M_t) \le k_t$$

which has a $\ell_1$-norm regularized convex objective function, but is subjecting to non-convex inequality rank constraints. A standard convexification of the rank constraints is to replace rank functions with their convex envelopes, nuclear norms [3, 14, 1, 6, 15]. For example, one can replace the rank constraints in (11) with relaxed nuclear norm regularizers in the objective function

$$\min_{M_c, M_s, M_t, E_s, E_t} \quad \frac{\alpha_s}{2} \|A_s M_c + M_s + E_s - X_s\|_F^2 + \frac{\alpha_t}{2} \|A_t M_c + M_t + E_t - X_t\|_F^2 \tag{12}$$

$$+ \beta_s \|E_s\|_1 + \beta_t \|E_t\|_1 + \lambda_c \|M_c\|_* + \lambda_s \|M_s\|_* + \lambda_t \|M_t\|_*$$

Many efficient and scalable algorithms have been proposed to solve such nuclear norm regularized convex optimization problems, including proximal gradient algorithm [6, 14], fixed point and Bregman iterative method [9]. However, though the nuclear norm is a convex envelope of the rank function, it is not always a high-quality approximation of the rank function [11]. Moreover, it is very difficult to select the appropriate trade-off parameters $\lambda_s, \lambda_t$ for the nuclear norm regularizers in (12) to recover the low-rank matrix solutions in the original optimization in (11). In principal component analysis problems it is much more convenient to have explicit control on the rank of the low-rank solution matrices. Therefore instead of solving the nuclear norm based convex optimization problem (12), we develop a scalable and efficient proximal gradient algorithm to solve the rank constraint based minimization problem (11) directly, which is shown to converge to a stationary point.

After solving the optimization problem (11), the low-rank approximation of the corrupted matrix $X_t$ can be obtained as $\hat{X}_t = A_t M_c + M_t$.

## 4  Proximal Projected Gradient Descent Algorithm

Proximal gradient methods have been popularly used for unconstrained convex optimization problems with continuous but non-smooth regularizers [2]. In this work, we develop a proximal projected gradient algorithm to solve the non-convex optimization problem with matrix rank constraints in (11). Let $\Theta = [M_c; M_s; M_t; E_s; E_t]$ be the optimization variable set of (11). We denote the objective function of (11) as $F(\Theta)$ such that

$$F(\Theta) = f(\Theta) + g(\Theta) \tag{13}$$

$$f(\Theta) = \frac{\alpha_s}{2} \|A_s M_c + M_s + E_s - X_s\|_F^2 + \frac{\alpha_t}{2} \|A_t M_c + M_t + E_t - X_t\|_F^2 \tag{14}$$

$$g(\Theta) = \beta_s \|E_s\|_1 + \beta_t \|E_t\|_1 \tag{15}$$

---

**Algorithm 1** Proximal Projected Gradient Descent

---

**Input:** data matrices $X_s, X_t$, parameters $\alpha_s, \alpha_t, \beta_s, \beta_t, k_c, k_s, k_t$.
**Set** $\eta = 3\max(\alpha_s, \alpha_t)$, $k = 1$.
**Initialize** $M_c^{(1)}, M_s^{(1)}, M_t^{(1)}, E_s^{(1)}, E_t^{(1)}$
**While** not converged **do**
- Set $\Theta^{(k)} = [M_c^{(k)}; M_s^{(k)}; M_t^{(k)}; E_s^{(k)}; E_t^{(k)}]$.
- Update $M_c^{(k+1)} = p_{M_c}(\eta, \Theta^{(k)})$, $M_s^{(k+1)} = p_{M_s}(\eta, \Theta^{(k)})$, $M_t^{(k+1)} = p_{M_t}(\eta, \Theta^{(k)})$,
  $E_s^{(k+1)} = p_{E_s}(\eta, \Theta^{(k)})$, $E_t^{(k+1)} = p_{E_t}(\eta, \Theta^{(k)})$.
- Set $k = k + 1$.
**End While**

---

Here $f(\Theta)$ is a convex and continuously differentiable function while $g(\Theta)$ is a convex but non-smooth function. For any $\eta > 0$, we consider the following quadratic approximation of $F(\Theta)$ at a given point $\widetilde{\Theta} = [\widetilde{M_c}; \widetilde{M_s}; \widetilde{M_t}; \widetilde{E_s}; \widetilde{E_t}]$

$$Q_\eta(\Theta, \widetilde{\Theta}) = f(\widetilde{\Theta}) + \langle \Theta - \widetilde{\Theta}, \nabla f(\widetilde{\Theta}) \rangle + \frac{\eta}{2}\|\Theta - \widetilde{\Theta}\|_F^2 + g(\Theta) \tag{16}$$

where $\nabla f(\widetilde{\Theta})$ is the gradient of the function $f(\cdot)$ at point $\widetilde{\Theta}$. Let $\mathcal{C} = \{\Theta : rank(M_c) \leq k_c, rank(M_s) \leq k_s, rank(M_t) \leq k_t\}$. The minimization over $Q_\eta(\Theta, \widetilde{\Theta})$ can be conducted as

$$p(\eta, \widetilde{\Theta}) = \arg\min_{\Theta \in \mathcal{C}} Q_\eta(\Theta, \widetilde{\Theta}) = \arg\min_{\Theta \in \mathcal{C}} \left\{ g(\Theta) + \frac{\eta}{2}\left\|\Theta - (\widetilde{\Theta} - \frac{1}{\eta}\nabla f(\widetilde{\Theta}))\right\|_F^2 \right\} \tag{17}$$

which admits the following closed-form solution through singular value decomposition and soft-thresholding:

$$p_{M_c}(\eta, \widetilde{\Theta}) = U_{k_c}\Sigma_{k_c}V_{k_c}^\top, \quad \text{for } U\Sigma V^\top = \text{SVD}(\widetilde{M_c} - \frac{1}{\eta}\nabla_{M_c}f(\widetilde{\Theta}))$$

$$p_{M_s}(\eta, \widetilde{\Theta}) = U_{k_s}\Sigma_{k_s}V_{k_s}^\top, \quad \text{for } U\Sigma V^\top = \text{SVD}(\widetilde{M_s} - \frac{1}{\eta}\nabla_{M_s}f(\widetilde{\Theta}))$$

$$p_{M_t}(\eta, \widetilde{\Theta}) = U_{k_t}\Sigma_{k_t}V_{k_t}^\top, \quad \text{for } U\Sigma V^\top = \text{SVD}(\widetilde{M_t} - \frac{1}{\eta}\nabla_{M_t}f(\widetilde{\Theta}))$$

$$p_{E_s}(\eta, \widetilde{\Theta}) = (|\widehat{E_s}| - \frac{\beta_s}{\eta})_+ \circ sign(\widehat{E_s}), \quad \text{with } \widehat{E_s} = \widetilde{E_s} - \frac{1}{\eta}\nabla_{E_s}f(\widetilde{\Theta})$$

$$p_{E_t}(\eta, \widetilde{\Theta}) = (|\widehat{E_t}| - \frac{\beta_t}{\eta})_+ \circ sign(\widehat{E_t}), \quad \text{with } \widehat{E_t} = \widetilde{E_t} - \frac{1}{\eta}\nabla_{E_t}f(\widetilde{\Theta})$$

where $U_k$ and $V_k$ denote the top $k$ singular vectors from $U$ and $V$ respectively, and $\Sigma_k$ denotes the diagonal matrix with the corresponding top $k$ singular values for $k \in \{k_c, k_s, k_t\}$ respectively; the operator "$\circ$" denotes matrix Hadamard product, and the operator $(\cdot)_+ = \max(\cdot, 0)$; $\nabla_{M_c}f(\widetilde{\Theta})$, $\nabla_{M_s}f(\widetilde{\Theta})$, $\nabla_{M_t}f(\widetilde{\Theta})$, $\nabla_{E_s}f(\widetilde{\Theta})$, and $\nabla_{E_t}f(\widetilde{\Theta})$ denote parts of the gradient matrix $\nabla f(\widetilde{\Theta})$ corresponding to $M_c, M_s, M_t, E_s, E_t$ respectively.

Our proximal projected gradient algorithm is an iterative procedure. After first initializing the parameter matrices to zeros, in each $k$-th iteration, it updates the model parameters by minimizing the approximation function $Q(\Theta, \Theta^{(k)})$ at the given point $\Theta^{(k)}$, using the closed-form update equations above. The overall procedure is given in Algorithm 1. Below we discuss the convergence property of this algorithm.

**Lemma 1** *For $\eta = 3\max(\alpha_s, \alpha_t)$, we have $F(\Theta) \leq Q_\eta(\Theta, \widetilde{\Theta})$ for every feasible $\Theta, \widetilde{\Theta}$.*

*Proof:* First it is easy to check that $\eta = 3\max(\alpha_s, \alpha_t)$ is a Lipschitz constant of $\nabla f(\Theta)$, such that

$$\|\nabla f(\Theta) - \nabla f(\widetilde{\Theta})\|_F \leq \eta\|\Theta - \widetilde{\Theta}\|_F \quad \text{for any feasible pair } \Theta, \widetilde{\Theta} \tag{18}$$

Thus $f(\cdot)$ is a continuously differentiable function with Lipschitz continuous gradient and Lipschitz constant $\eta$. Following [2, Lemma 2.1], we have

$$f(\Theta) \leq f(\widetilde{\Theta}) + \langle \Theta - \widetilde{\Theta}, \nabla f(\widetilde{\Theta}) \rangle + \frac{\eta}{2}\|\Theta - \widetilde{\Theta}\|_F^2 \quad \text{for any feasible pair } \Theta, \widetilde{\Theta} \tag{19}$$

Based on (16) and (19), we can then derive

$$F(\Theta) = f(\Theta) + g(\Theta) \le f(\widetilde{\Theta}) + \langle \Theta - \widetilde{\Theta}, \nabla f(\widetilde{\Theta}) \rangle + \frac{\eta}{2} \|\Theta - \widetilde{\Theta}\|_F^2 + g(\Theta) = Q_\eta(\Theta, \widetilde{\Theta}) \quad (20)$$

$\square$

Based on this lemma, we can see the update steps of Algorithm 1 satisfy

$$F(\Theta^{(k+1)}) \le Q_\eta(\Theta^{(k+1)}, \Theta^{(k)}) \le Q_\eta(\Theta^{(k)}, \Theta^{(k)}) = F(\Theta^{(k)}) \quad (21)$$

Therefore the sequence of points, $\Theta^{(1)}, \Theta^{(2)}, \ldots, \Theta^*$, produced by Algorithm 1 have nonincreasing function values $F(\Theta^{(1)}) \ge F(\Theta^{(2)}) \ge \ldots \ge F(\Theta^*)$, and converge to a stationary point.

## 5 Experiments

We evaluate the proposed approach using image denoising tasks constructed on the Yale Face Database, which contains 165 grayscale images of 15 individuals. There are 11 images per subject, one per different facial expression or configuration.

Our goal is to investigate the performance of the proposed approach on recovering data corrupted with large and dense errors. Thus we constructed noisy images by adding large errors. Let $X_t^0$ denote a target image matrix from one subject, which has values between 0 and 255. We randomly select a fraction of its pixels to add large errors to reach value 255, where the fraction of noisy pixels is controlled using a noise level parameter $\sigma$. The obtained noisy target image matrix is $X_t$. We then use an uncorrupted image matrix $X_s^0$ from the same or different subject as the source matrix to help the image denoising of $X_t$ by recovering its low-rank approximation matrix $\hat{X}_t$. In the experiments, we compared the performance of the following methods on image denoising with large errors:

- *R-T-PCA:* This is the proposed *robust transfer PCA* method. For this method, we used parameters $\alpha_s = \alpha_t = 1, \beta_s = \beta_t = 0.1$, unless otherwise specified.
- *R-S-PCA:* This is a *robust shared PCA* method that applies a rank-constrained version of the robust PCA in [14] on the concatenated matrix $[X_s^0; X_t]$ to recover a low-rank approximation matrix $\hat{X}_t$ with rank $k_c + k_t$.
- *R-PCA:* This is a *robust PCA* method that applies a rank-constrained version of the robust PCA in [14] on $X_t$ to recover a low-rank approximation matrix $\hat{X}_t$ with rank $k_c + k_t$.
- *S-PCA:* This is a *shared PCA* method that applies PCA on concatenated matrix $[X_s^0; X_t]$ to recover a low-rank approximation matrix $\hat{X}_t$ with rank $k_c + k_t$.
- *PCA:* This method applies PCA on the noisy target matrix $X_t$ to recover a low-rank approximation matrix $\hat{X}_t$ with rank $k_c + k_t$.
- *R-2Step-PCA:* This method exploits the auxiliary source matrix by first performing robust PCA over the concatenated matrix $[X_s^0; X_t]$ to produce a shared matrix $M_c$ with rank $k_c$, and then performing robust PCA over the residue matrix $(X_t - A_t M_c)$ to produce a matrix $M_t$ with rank $k_t$. The final low-rank approximation of $X_t$ is given by $\hat{X}_t = A_t M_c + M_t$.

All the methods are evaluated using the root mean square error (RMSE) between the true target image matrix $X_t^0$ and the low-rank approximation matrix $\hat{X}_t$ recovered from the noisy image matrix. Unless specified otherwise, we used $k_c = 8, k_s = 3, k_t = 3$ in all experiments.

### 5.1 Intra-Subject Experiments

We first conducted experiments by constructing 15 transfer tasks for the 15 subjects. Specifically, for each subject, we used the first image matrix as the target matrix and used each of the remaining 10 image matrices as the source matrix each time. For each source matrix, we repeated the experiments 5 times by randomly generating noisy target matrix using the procedure described above. Thus in total, for each experiment, we have results from 50 runs. The average denoising results in terms of root mean square error (RMSE) with noise level $\sigma = 5\%$ are reported in Table 1. The standard deviations for these results range between 0.001 and 0.015. We also present one visualization result for Task-1 in Figure 1. We can see that the proposed method *R-T-PCA* outperforms all other methods

Table 1: The average denoising results in terms of RMSE at noise level $\sigma = 5\%$.

| Tasks | R-T-PCA | R-S-PCA | R-PCA | S-PCA | PCA | R-2Step-PCA |
|--------|---------|---------|-------|-------|-------|-------------|
| Task-1 | 0.143 | 0.185 | 0.218 | 0.330 | 0.365 | 0.212 |
| Task-2 | 0.134 | 0.167 | 0.201 | 0.320 | 0.353 | 0.202 |
| Task-3 | 0.136 | 0.153 | 0.226 | 0.386 | 0.430 | 0.215 |
| Task-4 | 0.140 | 0.162 | 0.201 | 0.369 | 0.406 | 0.215 |
| Task-5 | 0.142 | 0.166 | 0.241 | 0.382 | 0.414 | 0.208 |
| Task-6 | 0.156 | 0.195 | 0.196 | 0.290 | 0.310 | 0.202 |
| Task-7 | 0.172 | 0.206 | 0.300 | 0.477 | 0.523 | 0.264 |
| Task-8 | 0.203 | 0.222 | 0.223 | 0.348 | 0.386 | 0.243 |
| Task-9 | 0.140 | 0.159 | 0.203 | 0.317 | 0.349 | 0.201 |
| Task-10 | 0.198 | 0.209 | 0.259 | 0.394 | 0.439 | 0.255 |
| Task-11 | 0.191 | 0.211 | 0.283 | 0.389 | 0.423 | 0.274 |
| Task-12 | 0.151 | 0.189 | 0.194 | 0.337 | 0.366 | 0.213 |
| Task-13 | 0.193 | 0.218 | 0.277 | 0.436 | 0.474 | 0.257 |
| Task-14 | 0.176 | 0.201 | 0.240 | 0.366 | 0.392 | 0.224 |
| Task-15 | 0.159 | 0.170 | 0.266 | 0.413 | 0.464 | 0.245 |

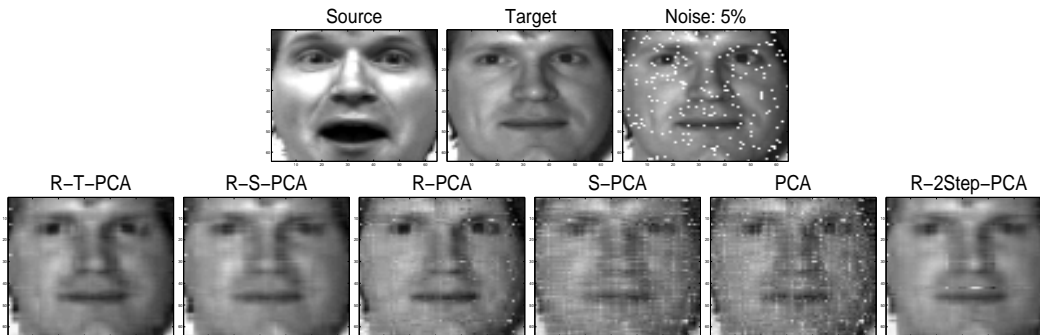

Figure 1: Denoising results on Task-1.

across all the 15 tasks. The comparison between the two groups of methods, {*R-T-PCA*, *R-S-PCA*, *S-PCA*} and {*R-PCA*, *PCA*}, shows that a related source matrix is indeed useful for denoising the target matrix. The superior performance of *R-T-PCA* over *R-2Step-PCA* demonstrates the effectiveness of our joint optimization framework over its stepwise alternative. The superior performance of *R-T-PCA* over *R-S-PCA* and *S-PCA* demonstrates the efficacy of our transfer PCA framework in exploiting the auxiliary source matrix over methods that simply concatenate the auxiliary source matrix and target matrix.

## 5.2 Cross-Subject Experiments

Next, we conducted transfer experiments using source matrix and target matrix from different subjects. We randomly constructed 5 transfer tasks, Task-6-1, Task-8-2, Task-9-4, Task-12-8 and Task-14-11, where the first number in the task name denotes the source subject index and second number denotes the target subject index. For example, to construct Task-6-1, we used the first image matrix from subject-6 as the source matrix and used the first image matrix from subject-1 as the target matrix. For each task, we conducted experiments with two different noise levels, 5% and 10%. We repeated each experiment 10 times using randomly generated noisy target matrix. The average results in terms of RMSE are reported in Table 2 with standard deviations less than $0.015$. We can see that with the increase of noise level, the performance of all methods degrades. But at each noise level, the comparison results are similar to what we observed in previous experiments: The proposed method outperforms all other methods. These results also suggest that even a remotely related source image can be useful. All these experiments demonstrate the efficacy of the proposed method in exploiting uncorrupted auxiliary data matrix for denoising target images corrupted with large errors.

Table 2: The average denoising results in terms of RMSE.

| Tasks | | R-T-PCA | R-S-PCA | R-PCA | S-PCA | PCA | R-2Step-PCA |
|---|---|---|---|---|---|---|---|
| Task-6-1 | $\sigma=5\%$ | 0.147 | 0.177 | 0.224 | 0.337 | 0.370 | 0.218 |
| | $\sigma=10\%$ | 0.203 | 0.246 | 0.326 | 0.490 | 0.526 | 0.291 |
| Task-8-2 | $\sigma=5\%$ | 0.132 | 0.159 | 0.234 | 0.313 | 0.354 | 0.200 |
| | $\sigma=10\%$ | 0.154 | 0.211 | 0.323 | 0.457 | 0.500 | 0.276 |
| Task-9-4 | $\sigma=5\%$ | 0.148 | 0.170 | 0.229 | 0.373 | 0.410 | 0.212 |
| | $\sigma=10\%$ | 0.204 | 0.240 | 0.344 | 0.546 | 0.585 | 0.282 |
| Task-12-8 | $\sigma=5\%$ | 0.207 | 0.231 | 0.245 | 0.373 | 0.397 | 0.249 |
| | $\sigma=10\%$ | 0.244 | 0.272 | 0.359 | 0.518 | 0.548 | 0.317 |
| Task-14-11 | $\sigma=5\%$ | 0.172 | 0.215 | 0.274 | 0.403 | 0.424 | 0.268 |
| | $\sigma=10\%$ | 0.319 | 0.368 | 0.431 | 0.592 | 0.612 | 0.372 |

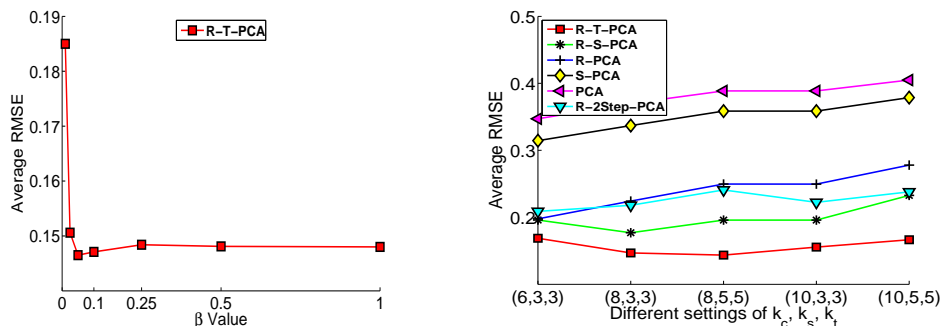

Figure 2: Parameter analysis on Task-6-1 with $\sigma = 5\%$.

## 5.3 Parameter Analysis

The optimization problem (11) for the proposed *R-T-PCA* method has a number of parameters to be set: $\alpha_s, \alpha_t, \beta_s, \beta_t, k_c, k_s$ and $k_t$. To investigate the influence of these parameters over the performance of the proposed method, we conducted two experiments using the first cross-subject task, Task-6-1, with noise level $\sigma = 5\%$. Given that the source and target matrices are similar in size, in these experiments we set $\alpha_s = \alpha_t = 1$, $\beta_s = \beta_t$ and $k_s = k_t$. In the first experiment, we set $(k_c, k_s, k_t) = (8, 3, 3)$ and study the performance of *R-T-PCA* with different $\beta_s = \beta_t = \beta$ values, for $\beta \in \{0.01, 0.025, 0.05, 0.1, 0.25, 0.5, 1\}$. The average RMSE results over 10 runs are presented in the left sub-figure of Figure 2. We can see that *R-T-PCA* is quite robust to $\beta$ within the range of values, $\{0.05, 0.1, 0.25, 0.5, 1\}$. In the second experiment, we fixed $\beta_s = \beta_t = 0.1$ and compared *R-T-PCA* with other methods across a few different settings of $(k_c, k_s, k_t)$, with $(k_c, k_s, k_t) \in \{(6, 3, 3), (8, 3, 3), (8, 5, 5), (10, 3, 3), (10, 5, 5)\}$. The average comparison results in terms of RMSE are presented in the right sub-figure of Figure 2. We can see that though the performance of all methods varies across different settings, *R-T-PCA* is less sensitive to the parameter changes comparing to the other methods and it produced the best result across different settings.

## 6 Conclusion

In this paper, we developed a novel robust transfer principal component analysis method to recover the low-rank representation of corrupted data by leveraging related uncorrupted auxiliary data. This robust transfer PCA framework combines aspects of both robust PCA and transfer learning methodologies. We formulated this method as a joint minimization problem over a convex combination of least squares losses with non-convex matrix rank constraints, and developed a proximal projected gradient descent algorithm to solve the proposed optimization problem, which permits a convenient closed-form solution for each proximal step based on singular value decomposition and converges to a stationary point. Our experiments over image denoising tasks demonstrated the proposed method can effectively exploit auxiliary uncorrupted image to recover images corrupted with random large errors and significantly outperform a number of comparison methods.

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
