[Reviews · NeurIPS 2013]

Submitted by Assigned_Reviewer_5

This paper considers robust principal component analysis, from the approach of transfer learning. The goal is to obtain a method that, according to the paper, can deal with not only small and/or sparse errors, but also dense large errors, in the setting where there are two data sources (two data matrices) which have some overlap in their principal components.

The authors then propose a rank-constrained optimization problem that is the natural formulation, assuming sparse errors; that is, they propose an objective which balances between the L2 loss in fitting the data matrix, plus an L1 penalty on the sparse corruption. This, in principle, should allow the handling of sparse noise, and also smaller dense noise. Instead of relaxing the rank constraints, they propose a projected proximal type iterative method, where they project back to matrices of appropriate rank, at every step.

There are several issues with this paper that if addressed, would significantly strengthen the contribution.

There are no conditions of success that are discussed. That is, the authors do not describe, theoretically, or even intuitively, what the conditions should be on the two data matrices and a the noise corrupting them, that would allow their method to succeed in finding the correct low dimensional structure. To be more specific, if we compare to the papers that the authors cite in the introduction, e.g., [11] or [14], or the numerous other papers that have been written on various formulations of robust PCA, e.g., papers by E. Candes et al., or papers by H. Xu et al., there, the authors give algorithms along with specific conditions for which they provide theoretical guarantees that their algorithms succeed. There does not seem to be something similar here. Precisely because it is not the usual setting, it would be very interesting to understand when transfer learning might succeed, where other methods might fail.

This is also very important because some of the claims the authors make do not seem to be backed up by a rigorous argument. The algorithm proposed is indeed natural. However, it is solving a non-convex problem, because of the rank constraints. By setting X_t or X_s to zero, it seems that one recovers the low-rank plus sparse recovery problem (see, e.g., Chandrasekaran et al., and Candes et al.). As mentioned previously, this is a well-studied and fairly well understood problem. Does their algorithm have better performance than the setting when the rank constraints are relaxed via the nuclear norm? The wording in the paper seems to suggest this, but there is nothing to support it.
Summary: The mathematical problem is interesting. However, the paper is missing some key points, including the setting for which their result would perform well, as well as the settings where their algorithm would succeed in solving the formulation they give.

Submitted by Assigned_Reviewer_6

The authors propose a new unsupervised dimensionality reduction method called the robust transfer PCA. A novelty of the proposed approach is that it clearly models the source and the target data. Moreover, they propose a convex formulation of the proposed approach which can be easily solved by using proximal projected gradient descent method. Through denoising experiments, the authors showed that the proposed method compares favorably with existing PCAs.

Detailed comments:
1. In experiments, what is the performance of robust PCA with only using source data (i.e., X_s^0)?
2. There are four parameters in the proposed method. I think this is sometime too many to tune.
Are there any heuristics to tune those parameters? Also, if you can add one or more experiments (not image denoising) with using the same parameter used in the denoising experiments, it would be nice.
3. This method may be able to use for unsupervised domain adaptation, since the problem setting in this paper is somewhat similar to the one in transfer component analysis (Pan et al., IJCAI2009). Thus, it is nice to include unsupervised domain adaptation in future.

Quality

This paper is technically sound.

Clarity

This paper is clearly written and well-organized.

Originality

The denoising problem is a very traditional problem.
The originality of the paper is that it clearly models source and target data in PCA.
Thus, I think this paper is somewhat original.

Significance

The proposed method outperforms existing methods.
In addition to denoising problems, the proposed method may be able to use for unsupervised domain adaptation problems.
Summary: The proposed problem setting is interesting. If authors can include more experimental results to show the robustness with respect to tuning parameters, it would be nice. The paper is well written and well-organized.

Submitted by Assigned_Reviewer_7

The method is novel and interesting, and the manuscript provides solid experimental proof to show the effectiveness of the method. The manuscript is overall well written.

Two concerns:
1. The manuscript neither gives a strategy how to choose the seven parameters in the proposed model nor shows the robustness of the model with regard to these parameters.
2. In P2, the formulation of PCA can be equation (1) when the data matrix X is assumed to be centered in columns, which is not claimed in the manuscript.
Summary: The manuscript proposed a novel robust transfer PCA which generalizes robust PCA in traditional machine learning setting to the transfer learning setting. A proximal projected gradient descent algorithm, which was shown to be convergent, is proposed to solve the optimization problem ,and the numerical experiments showed the effectiveness of the algorithm in applications.

Submitted by Meta_Reviewer_2

This paper proposed a robust transfer PCA formulation which generalizes robust PCA to the transfer learning setting. A proximal projected gradient descent algorithm is proposed to solve the optimization problem with guaranteed convergence. The main novelty is that it clearly models source and target data in RPCA. The paper is very well written.

How to tune the parameters in practice is a major challenge. The authors should provide some general guideline in the paper. Some discussions on when the proposed algorithm may fail will also be helpful.
Summary: This paper proposed a robust transfer PCA formulation which generalizes robust PCA to the transfer learning setting. The main novelty is that it clearly models source and target data in RPCA.
Author Feedback

Author rebuttal: 